



# Measurement report: Comparison of wintertime individual particles at ground level and above the mixed layer in urban Beijing

**Wenhua Wang[1, 2, 3], Longyi Shao[1*], Claudio Mazzoleni[3], Yaowei Li[1], Simone Kotthaus[4], Sue Grimmond[4], Janarjan Bhandari[3], Jiaoping Xing[1, 5], Xiaolei Feng[1], Mengyuan Zhang[1], Zongbo Shi[6]**

1. State Key Laboratory of Coal Resources and Safe Mining & College of Geosciences and Surveying Engineering, China University of Mining and Technology, Beijing, 100083, China

2. School of Resources and Materials, Northeastern University at Qinhuangdao, Qinhuangdao, 066004, China

3. Atmospheric Sciences Program & Physics Department, Michigan Technological University, Houghton, 49931, USA

4. Department of Meteorology, University of Reading, Reading, RG6 6BB, UK

5. School of Forestry, Jiangxi Agricultural University, Nanchang, 330045, China

6. School of Geography Earth and Environmental Sciences, the University of Birmingham, Birmingham, B15 2TT, UK

* Corresponding author: ShaoL@cumtb.edu.cn

**Abstract:**

Beijing has been suffering from frequent severe air pollution events, with concentrations affected significantly by the mixed layer height. Major efforts have been made to study the physico-chemical properties, composition, and sources of aerosol particles at ground level. However, little is known on morphology, elemental composition, and mixing state of aerosol particles above the mixed layer. In this work, we collected individual aerosol particles simultaneously at ground level (2 m above ground) and above the mixed layer in urban Beijing (within the Atmospheric Pollution and Human Health in a Chinese Megacity (APHH-Beijing) 2016 winter campaign). The particles were analyzed off-line using transmission electron microscopy coupled with energy dispersive X-ray spectroscopy. Our results showed that the relative number contribution of mineral particles to all measured particles was much higher during non-haze periods (42.5%) than haze periods (18.1%); on the contrary, internally mixed particles contributed more during haze periods (21.9%) than non-haze periods (7.2%) at ground level. In addition, more mineral particles were found at ground level than above the mixed layer height. Around 20% of individual particles showed core-shell structures during haze periods, whereas only a few core-shell particles were observed during non-haze periods (2%). We found that the particle above the mixed layer tend to be more aged with a larger proportion of organic particles originated from coal combustion. Our results indicate that a significant fraction of the airborne particles above the mixed layer originated from surrounding areas influenced by coal combustion activities. This source contributes to the surface particle concentrations in Beijing when polluted air is mixed down to the ground level.

**Keywords:** haze; individual particle; organic particle; core-shell structure; mixed layer.





**1. Introduction**
Atmospheric aerosols, emitted from anthropogenic or natural sources, consist of various
chemical constituents (e.g., organic matter, black carbon, nitrate, sulfate, ammonium, metals,
mineral dust) (Merikallio et al., 2011; Guo et al., 2014; Wang et al., 2016; Peng et al., 2016; Shao et
al., 2017; Tao et al., 2017). Anthropogenic aerosols have received increased attention in the last
decades due to their effects on climate and the environment. In fact, anthropogenic aerosols affect
climate through cloud condensation nuclei activity (Kerminen et al., 2012), hygroscopic growth
(Brock et al., 2016), and light scattering and absorption (Jacobson, 2001; Bond and Bergstrom, 2006;
Merikallio et al., 2011; China et al., 2013; Peng et al., 2016; Bhandari et al., 2019b). They can also
adversely impact human health; for example, by carrying toxic and carcinogenic compounds (Chen
et al., 2013; Shao et al., 2016, 2017). High concentrations of aerosol particles in urban air can cause
cardiovascular, respiratory, and even nervous system diseases (Xia et al., 2018; De Marco et al.,
2019; Shou et al., 2019). It is suggested that outdoor air pollution causes 3.3 million people
premature deaths globally each year (Lelieveld et al., 2015). Atmospheric aerosol particles also
affect regional and global geochemical cycles when they are transported over long distances (Heald
et al., 2006; Weijun Li et al., 2017; Rodriguez-Navarro et al., 2018).
Recently, China has suffered from severe air pollution conditions, like other countries
undergoing rapid social and economic development (Huang et al., 2014). In China, this has been
associated with frequent occurrence of haze episodes, high $PM_{2.5}$ mass levels, and expanded haze
areas (Guo et al., 2014; Huang et al., 2014; Sun et al., 2014). For example, the maximum hourly
average $PM_{2.5}$ mass concentrations reached more than 1000 μg m$^{-3}$ in Beijing winter time (Li et al.,
2017a; Zhang et al., 2017), 40 times above the safe level of 25 μg m$^{-3}$ recommended by the World
Health Organization (WHO).
As the megacity capital, Beijing has received much attention being one of the most polluted
cities in China. Atmospheric researchers have focused on aerosol particles to understand haze
formation in China (Sun et al., 2013; Huang et al., 2014; Zhou et al., 2018b). Measurements and
model analyses highlight the key roles of secondary aerosol formation by trace gases (e.g., volatile
organic compounds, and $SO_2$, $NO_x$) and stagnant meteorological conditions in the regional haze
formation (Wang et al., 2013; Guo et al., 2014; Huang et al., 2014).



As characterization of aerosol particles has focused on surface level observations, the
knowledge of aerosol properties at higher altitudes in urban areas remains poor (Zhou et al., 2018a).
Vertical differences between precursors, oxidants and temperature gradients may influence gas-
particle partitioning and heterogeneous reactions of $N_2O_5$ (Zhou et al., 2018a). Previous Beijing
measurements at the Institute of Atmospheric Physics (IAP) meteorological tower showed complex
vertical distributions of particulate matter and gaseous pollutants (Meng et al., 2008; Sun et al., 2015;
Wang et al., 2018; Zhou et al., 2018b), but most of these studies focused on non-refractory
submicron species. Research showed that the mixed layer height (MLH) could explain some of the
vertical difference of aerosol particle chemical composition (Sun et al., 2015; Wang et al., 2018).
For example, vertical distributions of aerosol particles tend to be more uniform during periods with
higher MLH (Wang et al., 2018). As heavily increased air pollution can reduce boundary layer
heights by diminishing incoming solar energy and therefore, by weakening vertical turbulence,
aerosol near-surface concentrations become elevated (Petaja et al., 2016). Moreover, the upper layer
particles can influence those below in the MLH by downward entrainment or mixing plumes,
making the lower layer particles more complex (Wehner et al., 2010; Platis et al., 2015; Qi et al.,
2019). Previous studies showed that the particles above the MLH sometimes considerably influence
cloud formation (Carnerero et al., 2018) and showed strong aerosol-radiation effect (Bond and
Bergstrom, 2006). The differences in aerosol types at ground level and at higher altitudes can lead
to large differences in aerosol direct forcing estimates (Ramanathan et al., 2002; Li et al., 2010).
The vertical difference of aerosol particles also increases the uncertainties in the assessment of the
climate system (Li et al., 2017b). Therefore, a detailed knowledge of the vertical distribution and
chemical composition of aerosols is important for understanding the impact on climate and the
aerosol evolution process (Zhang et al., 2009; Wang et al., 2018).
Vertical comparisons of individual aerosol particles and their morphology, mixing states, and
elemental compositions are very limited. Transmission Electron Microscopy (TEM) can provide
detailed individual particle characterization and help to explain heterogeneous reactions and aging
process (Li et al., 2016a). In this study, we compare particles simultaneously collected at ground
level and above the MLH based on the meteorological tower at IAP in Beijing, as part of the UK-
CHINA atmospheric pollution and Human Health (APHH) 2016 winter campaign.



## 2. Experimental

### 2.1. Aerosol sampling

Individual aerosol samples were collected at the tower division of IAP, Chinese Academy of Science (39°58′N, 116°22′E), in Beijing from 1 to 9 December of 2016. The site, located between the north $3^{rd}$ and $4^{th}$ ring roads in Beijing, is influenced by surrounding and regional traffic, and commercial, as well as, residential activities (Sun et al., 2016).

Two DKL-2 single stage cascade impactors, with a 0.5-mm-diameter jet nozzle and a flow rate of 1 L min$^{-1}$ were used. The sampler collection efficiency is ~ 100% at an aerodynamic diameter of 0.5 μm if the particle density is 2 g·cm$^{-3}$ (Li et al., 2016b). Copper (Cu) TEM grids, coated with carbon film (300-mesh; Tianld Co.; Beijing, China), were used to collect the aerosol samples. The sampling duration ranged from 30 second to less than 5 minutes depending on the air pollution loads. Simultaneous observation at ground level (Z1; 2 m above ground) and an elevated altitude (Z2; 280 m above ground) allowed us for the vertical profile of the characteristics of the particles to collected. The collected samples were stored in a dry plastic tube and placed in an air dryer to minimize particle changes before analysis.

Automatic Lidar and Ceilometer (ACL) observations of attenuated backscatter were conducted at the site using a Vaisala CL31 sensor. Measurements were corrected to account for instrument-related background and near range artefacts (Kotthaus et al., 2016). The MLH was derived from profile measurements using the automatic CABAM algorithm (Kotthaus and Grimmond, 2018). Since the TEM samples were collected for less than 5 minutes, the MLH at 15 min resolution was used to determine whether the Z2 observations were located within the MLH or above the MLH (Shi et al., 2019).

Samples were obtained during the periods shown (solid dots and dashed lines) in Fig. 1a; detailed sample information is provided in Table 1. Other measurements including PM$_{2.5}$, SO$_2$, NO$_2$, and O$_3$ mass concentrations at ground level were obtained from the Olympic Park monitor site, which is the closest national air quality monitor station to IAP (~1.5 km) (Shi et al., 2019). City average temperature (T) and relative humidity (RH) at ground level were obtained from the Ministry of Ecology and Environment of China (https://www.aqistudy.cn/).

### 2.2. Individual particle analysis





Individual particles were analyzed using a JEOL JEM-2800 TEM at an accelerating voltage
of 200 kV. The morphology and mixing state of individual particles were determined from the TEM
images. Semi-quantitative elemental composition was determined using Energy Dispersive X-ray
Spectroscopy (EDS), by which elements heavier than Boron (Z⩾6) can be detected. Cu is not
included in this paper because the TEM grids were made of copper. The aerosol particles were not
evenly distributed on the TEM grids; the coarser particles occurred near the center and the finer
particles occurred on the periphery. To ensure a representative data analysis, three or four areas from
the center to the periphery were selected and analyzed. The EDS collection duration of each
individual particle was about 15 s to reduce damage of particles from the electron beam. The
projected areas of individual particles were determined using the Image-J software (Schneider et al.,
2012), which is commonly used for counting and measuring the projected area of atmospheric
particles acquired by electron microscopes (Unga et al., 2018). First, the gray-scale images of the
particles were converted into binary images, in which black pixels represent the particles and white
pixels represent the background. The area equivalent diameters ($D_{Aeq}$) of the particles were
calculated by the following formula: $D_{Aeq}= 2 \cdot (A/\pi)^{1/2}$, where A is the projected area of the particles
shown in the TEM image.
**3.   Results and discussions**
**3.1.   Air pollutants mass concentrations**
The temporal variations of different air pollutants and meteorological conditions at ground
level are shown in Fig. 1. The hourly averaged $PM_{2.5}$ mass concentration at the Olympic Park
monitoring site ranged from 3 to 530 μg m$^{-3}$, with a sample period average of 113.3 μg m$^{-3}$,
significantly exceeding the safe level of 75 μg m$^{-3}$ according to the Chinese National Ambient Air
Quality Standard (GB3095-2012). The MLH ranged from 54 to 1496 m, with an average of 397 m.
The MLH showed obvious daily cycles. The hourly mean RH ranged from 17% to 97%, with a 9
day mean of 50.3%. The RH and $PM_{2.5}$ are positively correlated (correlation coefficient=0.75; Fig.
S1) according to the 216 groups of hourly data, suggesting that the higher RH favors the formation
of haze (Sun et al., 2014;Wang et al., 2016). As expected, RH and temperature were negatively
correlated (correlation coefficient=-0.51; Fig. S2). The $SO_2$ time series has similar trends to that of
$NO_2$. However, the average concentration of $NO_2$ (83.2 μg m$^{-3}$) was nearly 5.5 times higher than





that of $SO_2$ (15.2 µg m$^{-3}$). The concentration of $O_3$ showed a different trend compared with $NO_2$ and
$SO_2$ (Fig. 1), with a 9 days hourly mean concentration of 20 µg m$^{-3}$.
**3.2.   Classification and mixing state of individual particles**

Aerosol particles were classified using their morphologies and elemental compositions into

seven main types, namely: 1) organic particles (OPs), 2) sulfur-rich (S-rich) particles, 3) soot
particles, 4) mineral particles, 5) metal particles, 6) internally mixed organic and sulfur-rich particles
(OP-S), and 7) other mixed particles. The detailed characteristics of each particle type are shown in
Table 2.

OPs are mainly composed of C and O, usually with a small amount of Si, S, Cl, and K. OPs

are relatively stable under the electron beam irradiation. Based on the morphologies, OPs can be
further divided into spherical (Fig. 2a) and irregularly shaped (Fig. 2b). They were mainly from
combustion process of biomass and fossil fuel.

S-rich particles (Figs. 2c and 2d) are mainly composed of O, S, and N, and sometimes also

contain some amount of K. S-rich particles are beam-sensitive and volatilize under strong beam
irradiation. S-rich particles generally represent secondary inorganic components (e.g., $SO_4^{2+}$, $NO_3^-$
and $NH_4^+$) (Xu et al., 2019).

Soot particles are mainly composed of C, minor amount of O, and sometimes Si. Soot particles

consist of a number of C-dominated spherical monomers less than 100 nm in diameter (Figs. 2e and
2f) and can be easily identified under high-resolution TEM (Buseck et al., 2014; Bhandari et al.,
2017). Soot particles, stable under the electron beam, show chain-like or compact morphology in
the atmosphere (Sorensen et al., 2001; Adachi et al., 2007; China et al., 2013, 2015; Bhandari et al.,
2019a). Soot particles are mainly generated during incomplete combustion of biomass and fossil
fuel.

Metal particles (Figs. 2g and 2h) and Mineral particles (Fig. 2i) are stable under the beam

irradiation. Mineral particles are mostly irregularly shaped containing crustal elements (e.g., Si, Al,
Ca, Fe, Na, K, Mg, Ti, and S) in addition to O. They can be generated from windblown soil dust or
road dust. Metal particles are spherical or near spherical and are mainly composed of Fe, Zn, Mn,
Ti, and Pb. Metal particles ae mainly originating from industries, coal-fired power plants, and oil
refineries (Xu et al., 2019) or vehicle brakes (Hou et al., 2018).





Internally mixed particles (Figs. 2j-p) are particles with at least two of the above components.
They usually show relative larger diameter. We further classify them as internally mixed organic
and sulfur-rich particles (OP-S) (Figs. 2j-l), and other mixed particles (Figs. 2m-p).
**3.3.    Ground level haze and non-haze individual particle comparison**
Haze periods are defined as when the hourly average PM$_{2.5}$ mass concentration is more than
75 μg m$^{-3}$; the rest are defined as non-haze periods. A total of 1538 individual particles among 8
samples at ground level were analyzed based on the TEM results. The relative number percentage
(N(type i)/ N(total)*100 ) of particles in each sample was calculated, and the results are provided in
Table 3 and shown in Fig.3. During non-haze periods, the particles were composed of mineral
particles (42.5%), OPs (21.1%), S-rich particles (20.0%), soot particles (6.4%), other mixed
particles (5.6%), metal particles (2.83%), and OP-S (1.6%) in descending order. During haze periods,
the particles were composed of OPs (28.3%), S-rich particles (23.5%), mineral particles (18.1%),
OP-S (13.1%), other mixed particles (8.8%), soot particles (6.6%), and metal particles (1.7%) in
descending order.
The mineral particles are mainly from re-suspended road dust, soil dust, and construction dust
during non-desert transport dust episodes (Sun et al., 2006; Gao et al., 2016; Wang et al., 2017). The
relative number percentage of mineral particles was much higher during non-haze periods (42.5%)
than during haze periods (18.1%), as shown in Fig.3. However, the mixed particles including OP-S
and other mixed particles were much more abundant during haze periods (21.9%) than during non-
haze periods (7.2%), suggesting that there was more secondary aerosol formation during haze
periods. High secondary aerosol formation in winter in Beijing during the pollution periods was also
found in previous studies (Huang et al., 2014; Sun et al., 2016; Li et al., 2017a). Secondary aerosol
formation is expected since the RH during the haze periods were relatively higher than during non-
haze periods, as shown in Table 1 and Fig.1, which facilitated chemical reactions of gaseous
pollutants (Liu et al., 2016; Wang et al., 2016). Also, the average OPs and S-rich were higher during
haze periods than during non-haze periods.
**3.4.    Ground level and above the MLH individual particle comparison**
A total of 1519 individual particles from 8 samples above the MLH were analyzed. The results
are provided in Table 3 and shown in Fig. 3. During non-haze periods, the contribution of mineral
particles above the MLH (23.2%) was less than that at ground level (42.5%), but the S-rich and OPs
accounted for 30.7% and 27.3% above the MLH, respectively, fractions higher than those of 20.0%
and 21.1% at the ground level. During haze periods, the contribution of mineral particles above the
MLH (9.5%) was also lower than at ground level (18.1%). S-rich particles were also less abundant
above the MLH (16.4%) compared to the ground level (23.5%), which is different from the non-
haze periods. This may be because more S-rich particles above the MLH were mixed with other
particles, forming mixed particles. The mixed particles above the MLH were much higher than at
ground level, especially the OP-S particles (20.7% vs 13.1%). OPs above the MLH (34.8%) were
more abundant than at ground level (28.5%). Particles above the MLH were either transported from
the surrounding areas or from ground sources. In both cases, they were subject to atmospheric
process, leading to their aging.
**3.5.    Aging of particles**

During the aging process of aerosol particles, secondary species can coat pre-existing particles

(Li and Shao, 2009; Laskin et al., 2016; Li et al., 2016b; Niu et al., 2016; Tang et al., 2016; Chen et
al., 2017; Hou et al., 2018; Unga et al., 2018; Xu et al., 2019). Using high-resolution TEM images,
it is possible to identify the core-shell structure of particles (Li et al., 2016a). For example, Figs. 4a
and 4b show S-rich particles coated by secondary species. Figs. 4c and 4d show Ops that were
coated with secondary species. Figs. 4e-h show core-shell structured particles with some mixed
particle cores. In this study, we found that the core-shell structured particles accounted for 20%
during haze periods but only 2% during non-haze periods. Also, the average $D_{Aeq}$ of particles was
larger during haze periods than during non-haze periods as shown in Fig. S3. These results
confirmed that particles during haze periods underwent more extensive aging than during non-haze
periods.

The coating of atmospheric particles is often caused by aging mechanisms such as coagulation,

condensation, and heterogeneous chemical reactions (Kahnert, 2015; Müller et al., 2017). Fig. 5
shows low magnification images of particles at ground level and above the MLH. The core/shell
ratio (R), which is the ratio of the $D_{Aeq}$ of the core to the $D_{Aeq}$ of the whole particle including the
coating, has been used to evaluate the aging process of aerosol particles in different studies (Niu et
al., 2012, 2016; Hou et al., 2018). The value of R ranged from 0 to less than 1. A smaller R value



means the particles were more coated, thus were subjected to a more extensive degree of aging (Hou
et al., 2018). Because a high number percentage of core-shell structured particles were only found
during haze periods, we measured R of core-shell structured particles only during the haze periods
(including the samples 2, 4, 5, 6 and 7). Fig. 6a shows the R value of each samples during the haze
periods. The average R value above the MLH (0.54) was smaller than ground level (0.59). We can
see from Fig. 6a that all the samples showed a smaller average R value above the MLH compared
with those from the ground level. Additionally, the relative number percentage of core-shell
structured particles was higher above the MLH than at ground level, except for sample 4. These
findings indicate that the particles above the MLH were more aged than those at ground level.
**4.     Summary and Atmospheric implications**

Our results show that mineral particles represented a higher number percentage during non-

haze periods (42.5%) than during haze periods (18.1%), and mixed particles were more abundant in
haze periods (21.9%) than in non-haze periods (7.2%) at the ground level. In addition, more mineral
particles were found at ground level than above the MLH. Our results also show higher relative
number percentage of OPs both during non-haze (21.1%) and haze periods (28.3%) in winter
Beijing, compared with a tunnel environment (~5%), where the vehicle emissions were the main
pollution sources (Hou et al., 2018). Also, recent studies did not find abundant OPs in North China
during Spring and Summer (Yuan et al., 2015; Li et al., 2016b; Xu et al., 2019;). Instead, a larger
number percentage of OPs have been found in winter using electron microscopy in previous studies,
including an outflow of a haze plume in East Asia (Zhu et al., 2013), a coal-burning region in China's
Loess Plateau (Li et al., 2012), three sampling sites in North China Plain (Chen et al., 2017) and
urban and rural sites in Northeast China (Xu et al., 2017; Zhang et al., 2017). The results above
suggest that OPs account for a large number percentage of the particles in north China in winter.

Most of the OPs in our study were spherical or nearly spherical in shape according to the

projected images, suggesting that they were formed through cooling process after the biomass or
fossil fuel combustion pyrolysis products of volatile organic compounds were emitted into the
atmosphere (Wang et al., 2015; Chen et al., 2017; Zhang et al., 2017). These spherical or near
spherical OPs were considered to be brown carbon (Zhang et al., 2020). Brown carbon plays a
significant role in atmospheric shortwave absorption and can cause warming of the atmosphere

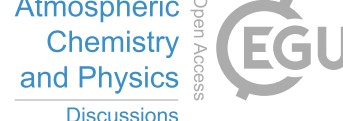

(Adachi and Buseck, 2011;Hoffer et al., 2016). Some researchers have found that the primary OPs
from coal combustion has more Si than those from biomass burning (Li et al., 2012; Chen et al.,
2017). The weight ratio of C-O-Si at ground level and above the MLH is shown in Fig. 7. More coal
burning related OPs were found above the MLH. Since the relative number percentage of primary
OPs affected by coal burning are higher above the MLH than at the ground level, the OPs above the
MLH are not all from the ground level and might have originated from surrounding areas influenced
by coal combustion. The particles above the MLH can contribute to Beijing air pollution by mixing
down to the ground.

In this study, more core-shell structured particles were found above the MLH than at the

ground; finding which can have important atmospheric implications. Fig. S3 shows the total particle
number-size distribution; the larger size particles clearly increased when considering the coatings
compared to only considering the core size during haze periods. The changes in optical properties
due to coating was calculated in various studies by using different methods (Cappa et al., 2012;
Scarnato et al., 2013; Liu et al., 2015; Saliba et al., 2016; Unga et al., 2018). When host particles
are coated, their optical properties might be amplified. For example, when soot particles (optically
often referred to as black carbon) are coated by secondary species, the light absorption is typically
enhanced because of the so called "lensing effect" (Khalizov et al., 2009; Liu et al., 2009; Peng et
al., 2016). Previous measurements showed that soot can heat the upper boundary layer more than
the lower layer during haze periods and decrease surface heat flux substantially, thus depressing the
development of MLH (Ding et al., 2016). Also, organic coating can influence the hygroscopic
properties and the viscosity of mixed particles (Sharma et al., 2018; Unga et al., 2018), and thus can
influence cloud formation activity (Kerminen et al., 2012).

The different relative number percentage of particle types and different aging degree of the

particles have important implications for understanding the climate effects of aerosol particles and
for emission control policy making.
**Data availability:** Data used in this study are available from the corresponding author upon request
(ShaoL@cumtb.edu.cn)
**Author Contributions:** WW, LS, CM, JX and ZS conceived the manuscript. WW, WL, XF and
MZ conducted the sample collection and analysis. SK and SG conducted the MLH measurement.
CM and BJ conducted manuscript modification.



**Competing interest:** The authors declare no conflict of interest.
**Acknowledgements:** We thank Zifa Wang and Pingqing Fu at IAP for the supporting of sample
collection. This work was supported by National Natural Science Foundation of China (No.
42075107), International Cooperation Projects of National Natural Science Foundation of China
(No. 41571130031), Yue Qi Scholar Fund of China University of Mining and Technology (Beijing),
China Scholarship Council (No. 201806430015). CM and JB were supported by the U.S Department
of Energy (DOE), Office of Biological and Environmental Research (OBER), Atmospheric System
Research (#DE-SC0011935 and Grant # DE-SC0018931). ZS was supported by Natural
Environmental Research Council (NE/N007190/1).



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




Table 1 Sample information and meteorological conditions

| Sample ID① | Date (2016) | Time ② | PM$_{2.5}$ (µg m$^{-3}$) | SO$_2$ (µg m$^{-3}$) | NO$_2$ (µg m$^{-3}$) | O$_3$ (µg m$^{-3}$) | RH (%) | T (°C) | MLH (m)③ |
|---|---|---|---|---|---|---|---|---|---|
| Z1-1 | 12/1 | 9:10 | 12④ | 2 | 48 | 37 | 24 | 6 | -- |
| Z2-1 | 12/1 | 8:40 | -- | -- | -- | -- | -- | -- | 194 |
| Z1-2 | 12/2 | 1:00 | 110 | 25 | 109 | 3 | 55 | 2 | -- |
| Z2-2 | 12/2 | 1:00 | -- | -- | -- | -- | -- | -- | 141 |
| Z1-3 | 12/2 | 9:10 | 24 | 20 | 134 | 2 | 50 | 3 | -- |
| Z2-3 | 12/2 | 8:40 | -- | -- | -- | -- | -- | -- | 134 |
| Z1-4 | 12/3 | 1:53 | 142 | 36 | 102 | 6 | 79 | -1 | -- |
| Z2-4 | 12/3 | 3:00 | -- | -- | -- | -- | -- | -- | 232 |
| Z1-5 | 12/4 | 1:04 | 530 | 14 | 180 | 4 | 93 | 1 | -- |
| Z2-5 | 12/4 | 3:00 | -- | -- | -- | -- | -- | -- | 136 |
| Z1-6 | 12/5 | 2:00 | 86 | 8 | 21 | 53 | 75 | 2 | -- |
| Z2-6 | 12/5 | 2:00 | -- | -- | -- | -- | -- | -- | 114 |
| Z1-7 | 12/8 | 9:10 | 187 | 2 | 16 | 72 | 86 | 2 | -- |
| Z2-7 | 12/8 | 8:40 | -- | -- | -- | -- | -- | -- | 191 |
| Z1-8 | 12/9 | 9:20 | 12 | 8 | 67 | 12 | 33 | 2 | -- |
| Z2-8 | 12/9 | 8:30 | -- | -- | -- | -- | -- | -- | 250 |

①Samples were collected at two altitudes: Z1 was 2 m above ground and Z2 was 280 m above
ground. ② Sampling duration ranged from 30 s to less than 5 minutes depending on the PM
pollution. ③ MLH represents the mixed layer height and the data are 15minutes average; MLH
was less than 280 m and the samples collected at Z2 represent samples above the mixed layer. ④
If PM$_{2.5}$ mass concentration was less than 75 µg m$^{-3}$, samples were classified as non-haze samples
and if PM$_{2.5}$ mass concentration was more than 75 µg m$^{-3}$, samples were classified as haze samples.



Table 2 Classification and characteristics of individual particle types.

| Particle type | Elemental composition | Morphology | Possible sources |
|---|---|---|---|
| Soot particles | C and minor amounts of O, Si. | Chain-like or compact C-dominated aggregates. | Incomplete combustion of biomass and fossil fuel. |
| Organic particles | C and O with minor amounts of Si, K, S, Cl. | Spherical, near spherical or irregular shapes. | Combustion process or secondary aerosol formation. |
| Mineral particles | O, Si, Al, Ca, Fe, Na, K, Mg, Ti, and S. | Irregular shapes. | Re-suspended from soil dust, road dust, and construction dust. |
| Metal particles | Fe, Zn, Mn, Ti, and Pb. | Spherical or irregular shapes. | Industries, coal-fired power plants and oil refineries. |
| S-rich particles | S and O with minor amounts of N, K. | Spherical, near spherical or irregular shapes. | Secondary aerosol formation. |
| Organic mixed with Sulfur-rich particles | C, O, and S with minor amounts of N, K or Cl. | Irregular shapes. | Secondary aerosol reaction. |
| Other mixed particles | Complex elemental composition. | Irregular shapes with different particle types. | Secondary aerosol reaction. |




Table 3 Relative number percentage of individual particles.

| Air qualities | Sample ID | Number | Metals | Minerals | OPs | S-rich | Soot | OP-S | Other |
|---|---|---|---|---|---|---|---|---|---|
| Non-haze periods | Z1-1 | 114 | 2.6 | 30.7 | 19.3 | 36.0 | 5.3 | 1.8 | 4.4 |
| | Z2-1 | 113 | 1.8 | 12.4 | 16.8 | 56.6 | 10.6 | 0.9 | 0.9 |
| | Z1-3 | 135 | 4.4 | 34.1 | 31.9 | 12.6 | 11.1 | 0.7 | 5.2 |
| | Z2-3 | 118 | 2.5 | 23.7 | 45.8 | 17.0 | 4.2 | 2.5 | 4.2 |
| | Z1-8 | 140 | 1.4 | 62.9 | 12.1 | 11.4 | 2.9 | 2.1 | 7.1 |
| | Z2-8 | 119 | 3.4 | 33.6 | 19.3 | 18.5 | 17.7 | 0.0 | 7.6 |
| | Ave (Z1) | 389 | 2.8 | 42.5 | 21.1 | 20.0 | 6.4 | 1.6 | 5.6 |
| | Ave (Z2) | 350 | 2.6 | 23.2 | 27.3 | 30.7 | 10.8 | 1.1 | 4.2 |
| Haze periods | Z1-2 | 123 | 2.4 | 21.1 | 42.3 | 17.1 | 7.3 | 2.4 | 7.3 |
| | Z2-2 | 164 | 4.9 | 14.6 | 37.2 | 25.0 | 4.3 | 9.8 | 4.3 |
| | Z1-4 | 160 | 0.6 | 28.8 | 30.6 | 8.8 | 13.8 | 9.4 | 8.1 |
| | Z2-4 | 266 | 0.0 | 3.8 | 53.0 | 3.4 | 7.1 | 19.6 | 13.2 |
| | Z1-5 | 461 | 0.9 | 6.5 | 18.9 | 22.1 | 7.6 | 31.5 | 12.6 |
| | Z2-5 | 266 | 0.4 | 0.4 | 32.3 | 7.1 | 2.3 | 44.0 | 13.5 |
| | Z1-6 | 237 | 2.5 | 11.0 | 21.5 | 48.5 | 2.1 | 6.8 | 7.6 |
| | Z2-6 | 281 | 1.8 | 11.0 | 18.9 | 19.6 | 12.8 | 15.3 | 20.6 |
| | Z1-7 | 168 | 1.8 | 23.2 | 28.0 | 20.8 | 2.4 | 15.5 | 8.3 |
| | Z2-7 | 192 | 1.6 | 17.7 | 32.3 | 27.1 | 1.6 | 15.1 | 4.7 |
| | Ave (Z1) | 1149 | 1.7 | 18.1 | 28.3 | 23.5 | 6.6 | 13.1 | 8.8 |
| | Ave (Z2) | 1169 | 1.7 | 9.5 | 34.7 | 16.4 | 5.6 | 20.7 | 11.3 |


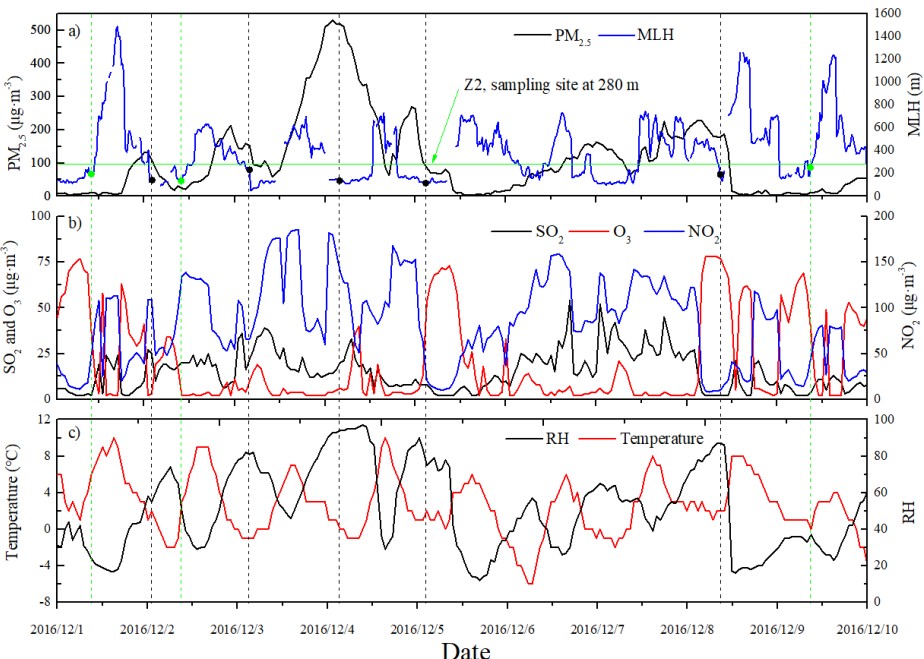

Fig. 1: The dashed lines represent the individual particle sampling times with green lines representing non-haze samples and black lines haze samples. (a) Temporal variations of mixed layer height (MLH) and $PM_{2.5}$ mass concentrations. The solid dots represent the MLH during the sampling times. (b) Temporal variations of $SO_2$, $NO_2$, $O_3$ at ground level at the Olympic Park monitor site, which is the closest national air quality monitor station to the sampling site (~1.5 km). (c) Temporal variations of temperature (T) and relative humidity (RH) at ground level. Date were obtained from the Ministry of Ecology and Environment of China (https://www.aqistudy.cn);

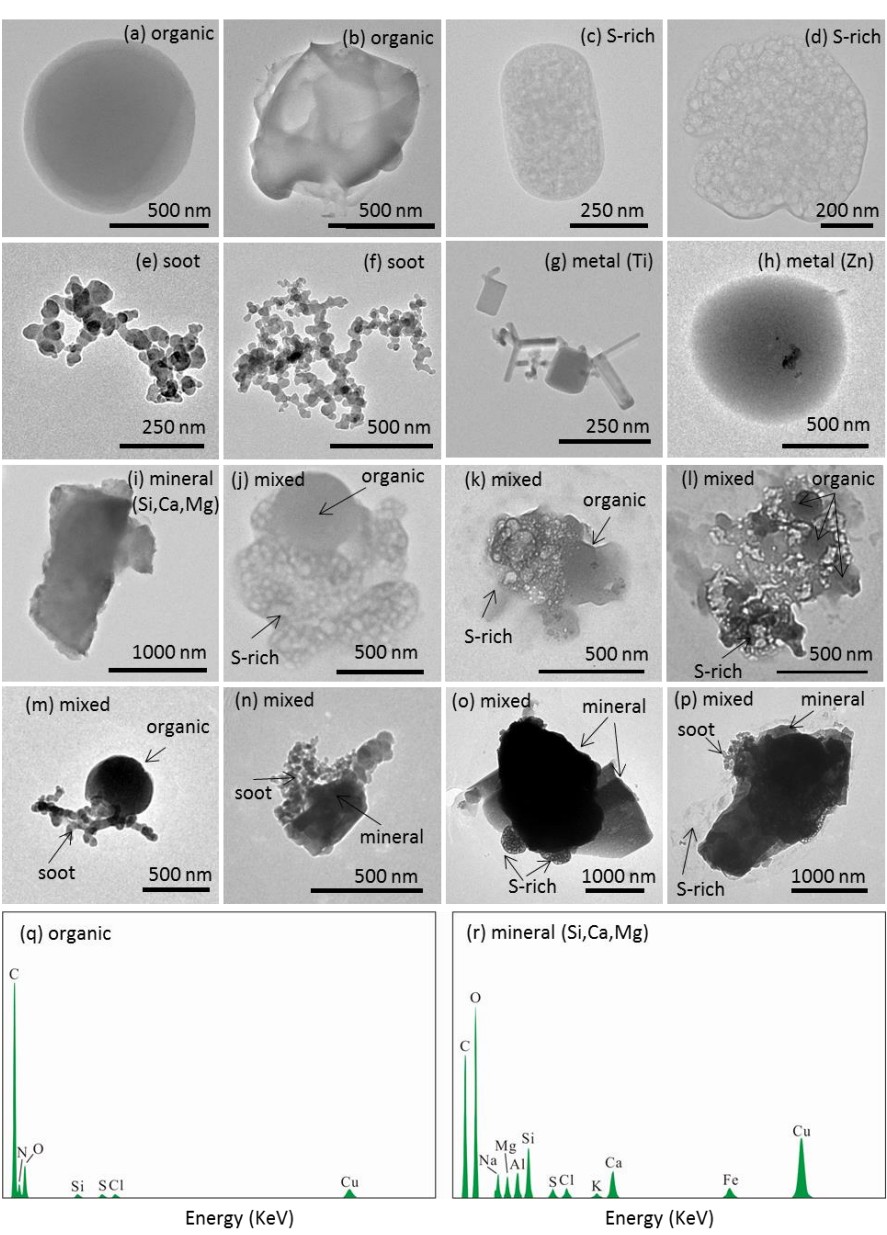

Fig. 2: Examples of morphologies and mixing characteristics of individual aerosol particles in winter in Beijing at ground level and above the mixed layer. (a) Spherical organic particle, (b) irregular shaped organic particle, (c-d) S-rich particles, (e-f) soot particles, (g-h) metal particles, (i) mineral particles, (j-l) OP-S mixed particles, and (m-p) other mixed particle types. (q) and (r) are EDS of (b) and (i). The difference between the particles in (b) and (i) is that organic particles (b) mainly composed C and O while minerals (i) mainly composed O, Si, Ca and Mg.




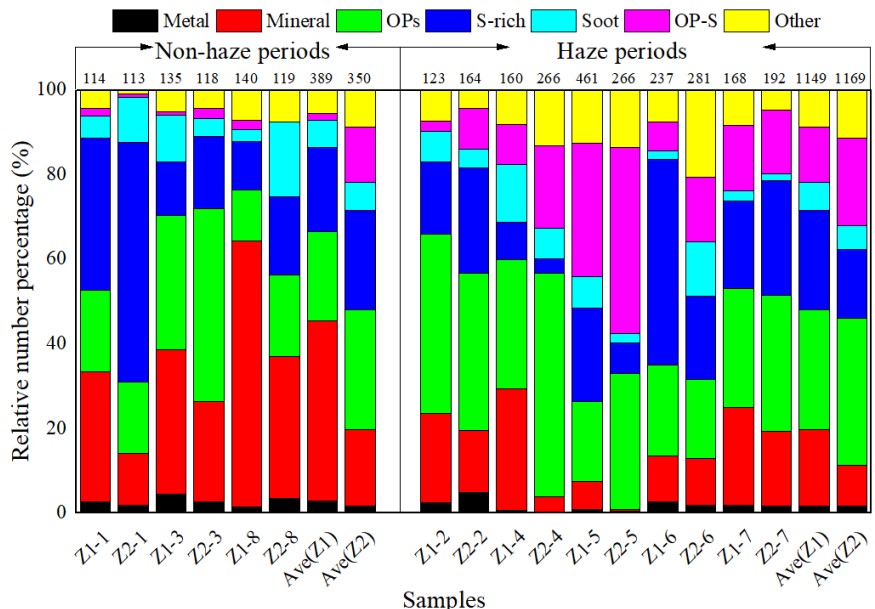


Fig. 3: Relative number percentage of different particle types at ground level (Z1) and above the
mixed layer height (Z2). The number above each bar represents the total particle number analyzed
in each sample.






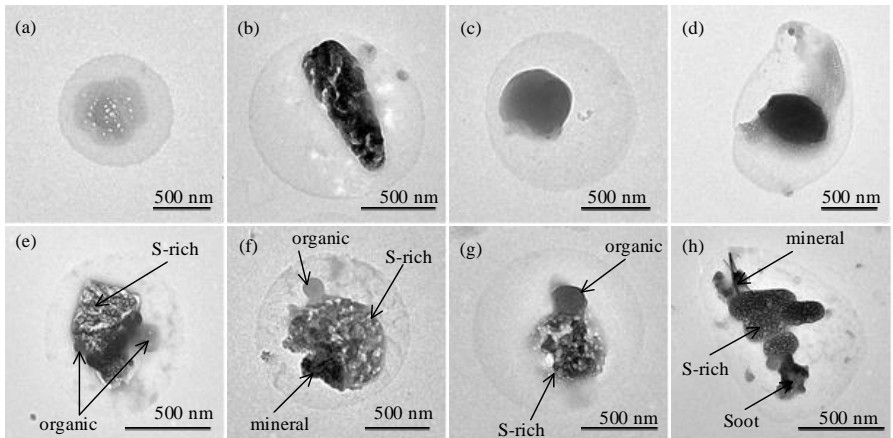


Fig. 4: Images of core-shell structured particles. (a-b) S-rich cores, (c-d) organic cores, and (e-h)
mixed cores.





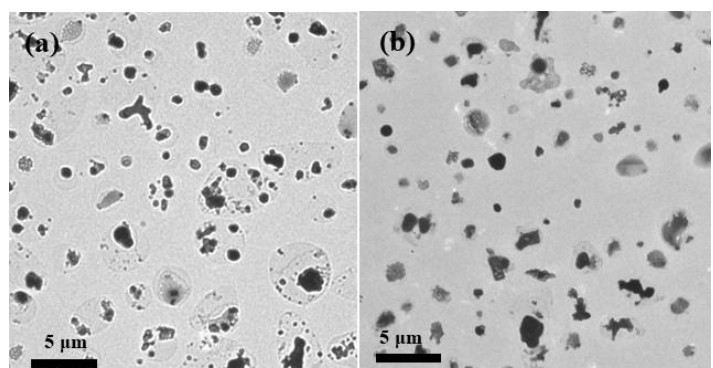


Fig. 5 Low magnification images of individual particles during haze periods above the MLH (a)
and at ground level (b). More coated particles are shown above the MLH.






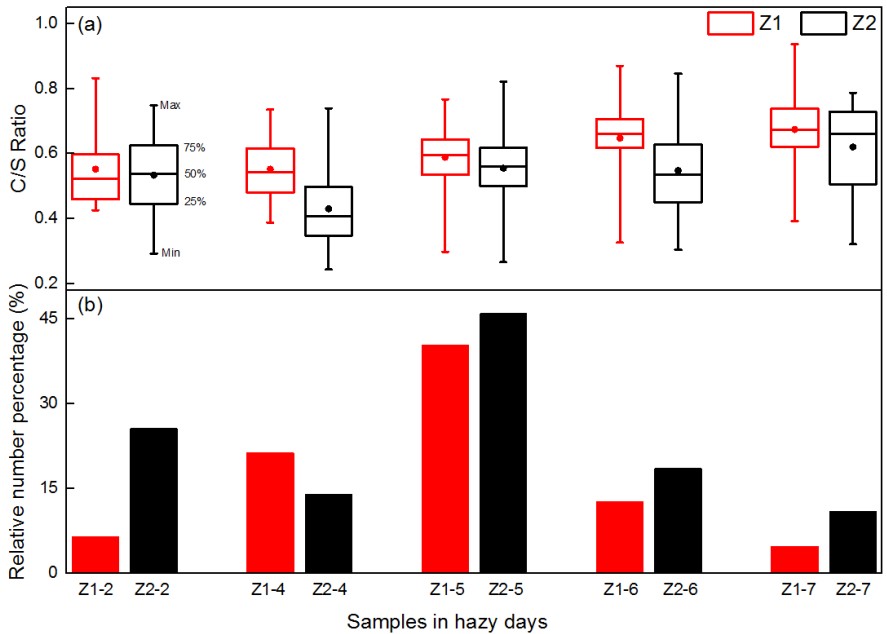

Fig. 6: (a) C/S ratio ($D_{Aeq}$ ratio of the core to the whole particle including the shell) of particles during haze periods at ground level (Z1) and above the mixed layer height (Z2); solid dots represent the average value, and (b) the corresponding relative number percentage of core-shell structured particles.





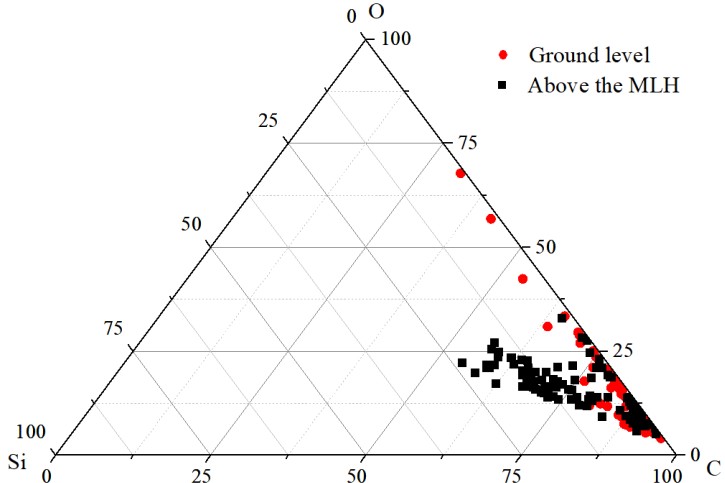

Fig. 7: Triangular diagram showing the weight ratio of C-O-Si of organic particles at ground level and above the mixed layer height (MLH).