# Peer review of "Measurement report: Comparison of wintertime individual particles at ground level and above the mixed layer in urban Beijing"

_Atmospheric Chemistry and Physics, 2020_

## Referee Comment (RC1) · Anonymous Referee #1 · 21 Dec 2020

This manuscript by Wang et al. shows the comparison of individual particles between the ground level and above mixing layer height (MLH) during wintertime in Beijing. This should be an interesting topic, however, after I read this paper, I did not get the significance of this study. In this study, the authors describe various particles in Beijing air. These particles are not surprising for me because they reported very details in cited papers. The authors shows the changes in number fractions and mixing state of individual particle types between non-haze and haze days at the ground level and between the ground level and above MLH. The paper was published as measurement report. As the low requirement for the paper innovation and quarlity, I might think that the paper can be published after one major revision.

[Figure]

(1) The fraction of mineral dust seems too high for me, what caused the high number fraction? The data is not right for normal haze event except the dust event. The fine secondary aerosols or primary particles should be dominant number in any case during clean and haze events. Obviously, the fundamental analysis might be not correct for individual particles. I supposed that the authors missed many fine particles in the TEM analysis.

(2) The lower magnification images should be provided to show differences. The authors didn't make notes in these two Figures. What are these aerosol particles? Could you add two low magnification images to show mineral particles?

(3) I noticed that the sampling time mainly at nighttime, when the MHL is the lowest. the authors missed samples at the daytime? Obviously, the potential readers are interested in the changes of particle types caused by the MLH change. Did the authors collected the samples in daytime? Then you can compare what differences when the MLH changed.

(4) If the authors can determine the particles above the MLH from the long-range transport or local surface emissions? More meteorological or models (e.g., HYSPLIT MODEL) should be added to indicate the particle transport.

(5) L236, the R value seem same between 0.54 and 0.59 including the errors. This value could be same. Also, I might think that the authors should add more transportation data here.

(6) Section 4, the implication should base on your own data. Seemly, some discussion or implication don't have any data support. The previous study should be not supporting all your discussion. Again, the authors should add more data to give more support for this part.

The paper has bad English writing. The authors should carefully revise it.

L138, Mass concentration of air pollutants.

L182 Comparison of haze and non-haze individual particle at ground level

L183-184, as could not connect one sentence

L188, OPs should change to OM (organic matter).

In this paper, there are many grammar mistakes. I didn't list all the english problem. The present and past states often mixed in one sentence.

---

## Referee Comment (RC2) · Anonymous Referee #2 · 30 Jan 2021

Review of "Measurement report: Comparison of wintertime individual particles at ground level and above the mixed layer in urban Beijing" by W. Wang, et al.

General comments

This manuscript presents work analyzing the characteristics of aerosol particles collected at two different heights (ground level and above the mixed layer height) in Beijing. The samples analyzed using TEM and EDS to generate both images of the particles as well as elemental composition. The samples are compared between two different types of environments: haze and non-haze. Classes of compounds that were identified include mineral, organic particles, soot particles, particles that contained sulfur (both

alone and mixed) as well as other types of internally mixed particles. The results from this study are interesting and the figures and discussion are clearly written. However, there are a few locations where more information or a clarification of statements would improve understanding. I would recommend acceptance after addressing the following minor concerns.

Specific Comments:

1. For the EDS analysis, were multiple places on the particles probed? It looks like they were based on the data in Figure 2. On page 5 line 128, you mention that the duration was 15 s to reduce damage. Please also include information in the methods on how many spots per particle were sampled.

2. For the area equivalent diameter calculations described on line 134 you are assuming particles are spherical. However, there are clearly non spherical particles in the figures. Please clarify what assumptions are being made and the uncertainty that can come from those assumptions in reporting the size distributions.

3. On page 6 lines 160-161 and also throughout the summary, statements are made about the organic particles (OP) coming from coal combustion or biomass and fossil fuel. How is this known? I agree that this could be a source, but how can you rule or other sources like secondary organic aerosol? Please clarify how this assignment of the source for these particles is being made.

4. In section 3.4 a comparison is made between samples at ground level and the samples collected on the tower. The averages are used for these comparisons, but some of the samples look like that trend is consistent across all the samples, and for others there is more variation. Please clarify where there is consistency and where there is more variation between the trends discussed for the average values.

5. On page 8, line 226: "In this study, we found that the core-shell structured particles accounted for 20% during haze periods but only 2% during non-haze periods". Was

this for the average of both heights? Or only one of the sample types (ground vs. tower)?

6. The area equivalent diameter is larger during haze periods (line 227-228). I can also see that more of the particles above the MLH have a thin coating surrounding them (Figure 5). However, the viscosity and the flattening behavior of these particles is not discussed. A more liquid-like coating will flatten more when the particle is impacted (collected), leading to a larger observed diameter on the substrate. This should be mentioned in the text and the effect of this on the calculated diameters and assumptions about coating thicknesses (amounts of aging in the manuscript) should be discussed. It is possible that the particles are actually quite similar in size, but that the physical properties of the films are different between the two heights. Was there a difference in RH during collection on the tower vs. at ground level? A different RH could mean more water associated with the particles and potentially a difference in viscosity for the coating.

7. Minor comments:

a. The green color on Figure 1 is hard to see

b. Add an explanation to the caption on Figure S3 to explain what it means to calculate the coating size; in the text it is discussed as an R value, but in Figure 6 it is the C/S ratio.

c. Are the times in Table 1 showing collection at 1 am (for example)? Or is that sup- posed to be 1 pm? Note, if these are at night, that should be much more clear in the text and you should discuss the implications (day vs. night, photochemistry, etc.).

d. Minor errors in grammar should be corrected to improve readability.

---

## Author Comment (AC1) · 11 Feb 2021

**Atmospheric Chemistry and Physics Manuscript ID acp-2020-1031**

**Reply to the Reviewer's comments**

Dear Editor,

We thank the reviewers for their insightful comments. We have revised our manuscript according to the suggestions. In the following section we give our answer (text in blue) to each of the points addressed by the reviewers. New text applied to meet the requests of the review is highlighted in red in the manuscript.

**Reviewer 1:**

(1) The fraction of mineral dust seems too high for me, what caused the high number fraction? The data is not right for normal haze event except the dust event. The fine secondary aerosols or primary particles should be dominant number in any case during clean and haze events. Obviously, the fundamental analysis might be not correct for individual particles. I supposed that the authors missed many fine particles in the TEM analysis.

Reply: The number percentage of the mineral particles is about 18.1%, which is normal for the haze samples in Beijing area (e.g., 25% in a previous study by Wang et al., 2015). The value is much lower than the mineral contents in the dust storm sample, with the latter typically higher than 80% (e.g., 90% in a previous study by Li et al., 2012). In addition, the particles analyzed in this study were mostly larger than 100 nm, we have added the description in line 144-145 in the manuscript).

Wenhua Wang, Longyi Shao, and Zexi Li et al., (2015). Morphologies and sulfation characteristics of individual aerosol particles in the haze episode over the Beijing-Tianjin-Tangshan area in January 2013. Acta Petrologica et Mineralogica. In Chinses with English Abstract.

Weijun Li and Longyi Shao. (2012). Chemical modification of dust particles during different dust storm episodes. Aerosol and Air Quality Research.

(2) The lower magnification images should be provided to show differences. The authors didn't make notes in these two Figures. What are these aerosol particles? Could you add two low magnification images to show mineral particles.

Reply: We have added notes in these figures and showed the comparison of mineral particles.

[Figure]

Fig. 5: Low magnification images of individual particles. (a) and (c) are particles above the mixed layer (MLH) at different size ranges. (b) and (d) are particles at ground level at different size ranges. More coated particles were found above the MLH. Arrows show part of the mineral particles.

(3) I noticed that the sampling time mainly at nighttime, when the MLH is the lowest. The authors missed samples at the daytime? Obviously, the potential readers are interested in the changes of particle types caused by the MLH change. Did the authors collect the samples in daytime? Then you can compare what differences when the MLH changed.

Reply: In this study, the particles were all collected in the morning and midnight when the MLH was the lowest and the height of the tower can reach the MLH at that time. Therefore, we can compare the particles at ground level and above the MLH. We have added some sentences in the manuscript in sample collection part. Please see line 121-123.

(4) If the authors can determine the particles above the MLH from the long-range transport or local surface emissions? More meteorological or models (e.g., HYSPLIT MODEL) should be added to indicate the particle transport.

Reply: we added a figure in the supplementary materials to show the long-range transport of particles. Air masses during haze periods in this study at 500 m height were mainly from the north and west. Please see line 297-299 and Fig. S4.

(5) L236, the R value seem same between 0.54 and 0.59 including the errors. This value could be same. Also, I might think that the authors should add more transportation data here.

Reply: We can clearly see in Fig. 6 that the value above the MLH was lower than at ground level. The average value 0.59 had a variance of 0.010 at ground level; the average value 0.54 had a variance of 0.015 above the MLH. There is a clear delineation between these two values. We have added a sentence to describe the transportation data. Please see line 98-99.

| Anova: Single Factor | | | | | | |
| --- | --- | --- | --- | --- | --- | --- |
| | | | | | | |
| SUMMARY | | | | | | |
| Groups | Count | Sum | Average | Variance | | |
| Column 1 | 266 | 157.543 | 0.59227 | 0.00998 | | |
| Column 2 | 272 | 146.593 | 0.53894 | 0.01462 | | |
| | | | | | | |
| | | | | | | |
| ANOVA | | | | | | |
| Source of Variation | SS | df | MS | F | P-value | F crit |
| Between Groups | 0.38238 | 1 | 0.38238 | 31.0144 | 4.1E-08 | 6.68239 |
| Within Groups | 6.6085 | 536 | 0.01233 | | | |
| | | | | | | |
| Total | 6.99088 | 537 | | | | |

Table 1 shows that F value is larger than F crit at the significance level of 0.01.

(6) Section 4, the implication should base on your own data. Seemly, some discussion or implication don't have any data support. The previous study should be not supporting all your discussion. Again, the authors should add more data to give more support for this part.

Reply: Many thanks for this comment. Section 4 (summary and atmospheric implications) has been spilt into two parts, including 3.6 possible sources of organic particles and 4 conclusions. Part of the sentences have been placed to the part 3.5 aging of the particles. We cited some papers to support that coated particles can have important implications for the atmosphere and hope to attract more researcher's attention (Please see line 266-271). We have shorted these sentences of the citied data in the manuscript.

The paper has bad English writing. The authors should carefully revise it
Line 138, mass concentration of air pollutants
Reply: Changed. See line 147.

L182 Comparison of haze and non-haze individual particle at ground level
Reply: Changed. See line 190.

L183-184, as could not connect one sentence
Reply: Changed. See line 191-192.

L188, OPs should change to OM (organic matter).
Reply: Organic matter (OM) are all the organic materials in the atmospheric aerosol. It is always used to calculated the weight of organic aerosol. Organic particles (OPs) are used by number. We think OPs might be more more appropriate.

In this paper, there are many grammar mistakes. I didn't list all the English problem.
The present and past states often mixed in one sentence.
Reply: We have carefully changed the English grammar.

---

## Author Comment (AC2)

**Atmospheric Chemistry and Physics Manuscript ID acp-2020-1031**

**Reply to the Reviewer's comments**

Dear Editor,

We thank the reviewers for their insightful comments and positive feedback. We have revised our manuscript according to the suggestions. In the following section we give our answer (text in blue) to each of the points addressed by the reviewers. New text applied to meet the requests of the review is highlighted in red in the manuscript.

**Reviewer 2:**

This manuscript presents work analyzing the characteristics of aerosol particles collected at two different heights (ground level and above the mixed layer height) in Beijing. The samples analyzed using TEM and EDS to generate both images of the particles as well as elemental composition. The samples are compared between two different types of environments: haze and non-haze. Classes of compounds that were identified include mineral, organic particles, soot particles, particles that contained sulfur (both alone and mixed) as well as other types of internally mixed particles. The results from this study are interesting and the figures and discussion are clearly written. However, there are a few locations where more information or a clarification of statements would improve understanding. I would recommend acceptance after addressing the following minor concerns.

Specific Comments:

1. For the EDS analysis, were multiple places on the particles probed? It looks like they were based on the data in Figure 2. On page 5 line 128, you mention that the duration was 15 s to reduce damage. Please also include information in the methods on how many spots per particle were sampled.

Reply: For most of the particles, we only collected one spectrum and we adjusted the spot size of beam according to the size of the particles. Therefore, we obtained the average elemental compositions of each particles. However, more than one spots per particles were sampled if the particles were inhomogeneous particles according to the TEM images. We have added some sentences in the methods part. Please line 130-134 in the manuscript.

2. For the area equivalent diameter calculations described on line 134 you are assuming particles are spherical. However, there are clearly non spherical particles in the figures. Please clarify what assumptions are being made and the uncertainty that can come from those assumptions in reporting

the size distributions.

Reply: Yes, there are non-spherical particles in the atmosphere. We cannot acquire the aerodynamic diameter distribution of particles according to the micrograph. There are some methods to calculate the diameter of particles, for example, the average of length and width of a bounding rectangle of the measured particles and area equivalent diameter. we calculated the area equivalent diameter assuming particles are spherical. These assumptions have been used in many previous microscopic studies as below:

(1) Zihan Wang, Wei Hu, and Hongya Niu., et al. (2021). Variations in physicochemical properties of airborne particles during a heavy haze-to-dust episode in Beijing. *Science of the Total environment.*

(2) Liang Xu, Daizhou Zhang and Weijun Li. (2019). Microscopic comparison of aerosol particles collected at an urban site in North China and a coastal site in Japan. *Science of the Total Environment*.

(3) Florin Unga, Marie Choël, and Yevgeny Derimian1., (2018). Microscopic observations of core-shell particle structure and implications for atmospheric aerosol remote sensing. *Journal of Geophysical Research: Atmospheric.*

(4) Janarjan Bhandari, Swarp China, and Kamal Kant., (2019). Extensive soot compaction by cloud processing from laboratory and field observations. *Scientific Report*.

(5) Wenhua Wang, Longyi Shao, and Jiaoping Xing., (2018). Physicochemical characteristics of individual aerosol particles during the 2015 China Victory Day Parade in Beijing. *Atmosphere*.

(6) Zongbo Shi, Longyi Shao, and T.P. Jones., (2003). Characterization of airborne individual particles collected in an urban area, a satellite city and a clean air area in Beijing, 2001. *Atmospheric Environment.*

3. On page 6 lines 160-161 and also throughout the summary, statements are made about the organic particles (OP) coming from coal combustion or biomass and fossil fuel. How is this known? I agree that this could be a source, but how can you rule or other sources like secondary organic aerosol? Please clarify how this assignment of the source for these particles is being made.

Reply: We admit that OPs might have different sources, but the combustion of coal or biomass and fossil fuel are the major sources in Beijing, which has been summarized by Li et al., (2016). We have rearranged the sentences to emphasize that OPs are mainly from coal combustion or biomass and fossil fuel. Please see line 169-170

Firstly, these spherical organic particles (OP) are stable under strong electron beam irradiation,

which is different from the secondary aerosols. Secondly, they appear dark features in the TEM images, which reflect their high thickness and refractory. Therefore, this kind of OP are primary particles instead of secondary aerosols. Please see line 283-288.

Li, W., Shao, L., Zhang, D., Ro, C.-U., Hu, M., Bi, X., Geng, H., Matsuki, A., Niu, H., and Chen, J.: A review of single aerosol particle studies in the atmosphere of East Asia: morphology, mixing state, source, and heterogeneous reactions, Journal of Cleaner Production, 112, 1330-1349, 10.1016/j.jclepro.2015.04.050, 2016a.

4. In section 3.4 a comparison is made between samples at ground level and the samples collected on the tower. The averages are used for these comparisons, but some of the samples look like that trend is consistent across all the samples, and for others there is more variation. Please clarify where there is consistency and where there is more variation between the trends discussed for the average values.

Reply: We have rearranged the sentences. Please see line below in red and line 215-231.

We found that the relative number percentage of mineral particles at ground level was larger than that above the MLH. For example, mineral particles at ground level and above the MLH during non-haze periods accounted for 42.5% and 23.2%, respectively, and during haze periods the values are 18.1% and 9.5%, respectively. S-rich particles during non-haze periods accounted for 20.0% at ground level, less than the value of 30.7% above the MLH. However, not all the samples above the MLH during haze periods showed higher relative number percentage of S-rich particles than at ground level. This might because some of the S-rich particles above the MLH were mixed with other particles, forming mixed particles. Another reason might be that higher relative number percentage of mixed particles diluted the relative number percentage of S-rich particles. The mixed particles during haze periods accounted for 32.0% above the MLH, higher than that of 21.9% at ground level. We also found that OPs above the MLH accounted for higher relative number percentage than at ground level, although there was some variance. For example, samples 4 and 6 showed higher relative number percentage of OPs at ground level. That might because that some of the OPs were mixed with S-rich particles and OP-S showed higher relative number percentage above the MLH than at ground level in samples 4 and 6. Metals and soot only accounted for a few relative number percentages in all samples and they didn't show much difference at ground level and above the MLH.

5. On page 8, line 226: "In this study, we found that the core-shell structured particles accounted for 20% during haze periods but only 2% during non-haze periods". Was this for the average of both heights? Or only one of the sample types (ground vs. tower)?

Reply: The value is the average of both heights. We emphasize that more core-shell particles were found during haze periods. We have added some sentence to show more details about the relative number percentage of core-shell particles. Please see below in red and line 241-245.

In this study, we found that the core-shell structured particles accounted for 20% during haze periods with 17% at ground level and 23% above the MLH, but only 2% during non-haze periods. These results demonstrated a general trend that the core-shell structured particles during haze periods were much higher than during non-haze periods.

6. The area equivalent diameter is larger during haze periods (line 227-228). I can also see that more of the particles above the MLH have a thin coating surrounding them (Figure 5). However, the viscosity and the flattening behavior of these particles is not discussed. A more liquid-like coating will flatten more when the particle is impacted (collected), leading to a larger observed diameter on the substrate. This should be mentioned in the text and the effect of this on the calculated diameters and assumptions about coating thicknesses (amounts of aging in the manuscript) should be discussed. It is possible that the particles are actually quite similar in size, but that the physical properties of the films are different between the two heights. Was there a difference in RH during collection on the tower vs. at ground level? A different RH could mean more water associated with the particles and potentially a difference in viscosity for the coating.

Reply: We admit that the particles collected by impaction may potentially be flattened. The absolute diameter of the flattened particles may be different from the diameter of the particle in the atmosphere. However, in this study, we compared the size distribution of particles before and after coating and emphasize that the particle size will increase when the particles are coated. We did not find obvious size difference between ground level and above the MLH.

7. Minor comments:

a. The green color on Figure 1 is hard to see

Reply: Changed.

b. Add an explanation to the caption on Figure S3 to explain what it means to calculate the coating size.

Reply: We have added an explanation to the caption on Fig. S3.

In the text it is discussed as an R value, but in Figure 6 it is the C/S ratio.

Reply: we replaced the C/S ratio with the R value throughout the entire manuscript.

c. Are the times in Table 1 showing collection at 1 am (for example)? Or is that supposed to be 1 pm? Note, if these are at night, that should be much clearer in the text and you should discuss the implications (day vs. night, photochemistry, etc.).

Reply: In this study, the particles were all collected in the morning and midnight when the MLH was the lowest and the height of the tower can reach the MLH. Therefore, we can compare the particles at ground level and above the MLH. We have added some sentences in the manuscript in sample collection part. Please see line 121-123.

d. Minor errors in grammar should be corrected to improve readability.

Reply: We have carefully changed the English grammar.

---

## Author Response (AR2)

**Reply to the Reviewer's comments**

Dear Editor,

We thank the reviewer for the insightful comments and positive feedback. We have revised our manuscript according to the suggestion.

The authors need to carefully use the OPs because they include the primary and secondary organic particles. These both particles have totally different morphology. The last part of in this study, I might suggest that the authors think about the primary organic particles. In one recent study, the authors found lots of burning-related particles: spherical organic particles which have been identified by tar ball in Beijing hazes (Persistent residential burning-related primary organic particles during wintertime hazes in North China: insights into their aging and optical changes. Atmos. Chem. Phys. 2021, 21, (3), 2251-2265. These particles can be important marker to represent the coal burning instead you mentioned all the OPs. If the authors can carefully revise part, the paper can be fully accepted for publication.

Reply: Many thanks. We replaced organic particles (OPs) with primary organic aerosols (POA) in the manuscript. We also cited the paper mentioned above.